

# Measuring recognition memory in zebrafish larvae: issues and limitations

Matteo Bruzzone[1,*], Elia Gatto[1,*], Tyrone Lucon Xiccato[2], Luisa Dalla Valle[3], Camilla Maria Fontana[3], Giacomo Meneghetti[3] and Angelo Bisazza[1,4]

[1] Department of General Psychology, University of Padova, Padova, Italy
[2] Department of Life Sciences and Biotechnology, University of Ferrara, Ferrara, Italy
[3] Department of Biology, University of Padova, Padova, Italy
[4] Padova Neuroscience Center, University of Padova, Padova, Italy
[*] These authors contributed equally to this work.

Corresponding author
Elia Gatto, elia.gatto@phd.unipd.it

## ABSTRACT

Recognition memory is the capacity to recognize previously encountered objects, events or places. This ability is crucial for many fitness-related activities, and it appears very early in the development of several species. In the laboratory, recognition memory is most often investigated using the novel object recognition test (NORt), which exploits the tendency of most vertebrates to explore novel objects over familiar ones. Despite that the use of larval zebrafish is rapidly increasing in research on brain, cognition and neuropathologies, it is unknown whether larvae possess recognition memory and whether the NORt can be used to assess it. Here, we tested a NOR procedure in zebrafish larvae of 7-, 14- and 21-days post-fertilization (dpf) to investigate when recognition memory first appears during ontogeny. Overall, we found that larvae explored a novel stimulus longer than a familiar one. This response was fully significant only for 14-dpf larvae. A control experiment evidenced that larvae become neophobic at 21-dpf, which may explain the poor performance at this age. The preference for the novel stimulus was also affected by the type of stimulus, being significant with tri-dimensional objects varying in shape and bi-dimensional geometrical figures but not with objects differing in colour. Further analyses suggest that lack of effect for objects with different colours was due to spontaneous preference for one colour. This study highlights the presence of recognition memory in zebrafish larvae but also revealed non-cognitive factors that may hinder the application of NORt paradigms in the early developmental stages of zebrafish.

## INTRODUCTION

The ability to learn and memorize the characteristics of objects and to recognize an object when it is encountered again is of fundamental importance for many fitness-related activities, such as obtaining food, avoiding predators or interacting with conspecifics. Recognition memory allows one to discriminate familiar stimuli from novel ones and to adjust one's behaviour accordingly (*Antunes & Biala, 2012*; *Blaser & Heyser, 2015*). In mammals and birds, recognition memory seems to appear very early. Researchers have observed recognition memory in pre-weaning rats, and they have reported the preference

for novel objects as early as the first week of an infant's life (*Johnson & Horn, 1986*; *Pascalis, 1994*; *Reger, Hovda & Giza, 2009*). The same pattern may occur in fish as well. Five-day-old guppies (*Poecilia reticulata*) can discriminate a familiar object from an unfamiliar one (*Miletto Petrazzini et al., 2012*). However, the reproductive mode of the guppy is atypical for fishes, as it is a livebearer. At birth, the fry is fully developed and endowed with a complex behavioural repertoire and a suite of cognitive abilities not much different from those of adults (*Magurran & Seghers, 1990*; *Bisazza et al., 2010*; *Miletto Petrazzini et al., 2012*; *Piffer, Miletto Petrazzini & Agrillo, 2013*).

Many other species of fish do not present such an advanced stage of development at birth. For example, the zebrafish (*Danio rerio*), which is the main teleost model for translational research, has a very short embryonal development and hatches at 2–3-days after fertilization. The nervous system of zebrafish larvae is poorly developed (*Nusslein-Volhard & Dahm, 2002*). They can swim, but they do not feed autonomously until the sixth-days post fertilization (dpf) and do develop some form of social behaviour only from 21 dpf (*Wilson, 2012*; *Hinz & De Polavieja, 2017*). There is little information available on the cognitive abilities of larval zebrafish. Valente and colleagues (*2012*) showed that zebrafish larvae can be conditioned to avoid one side of the tank by repeatedly pairing this position with electroshock. Learning started from approximately 21 dpf, and adult performance was reached around 42 dpf. Another study demonstrated the possibility of conditioning the tail contraction in 6- to 8-dpf zebrafish after repeatedly pairing a moving spot of light with a touch of the larva's body (*Aizenberg & Schuman, 2011*). However, it is unknown whether zebrafish larvae memorize the features of the objects that they encounter and whether they can discriminate amongst different objects.

Memory research with animal models was greatly boosted by the introduction of a new paradigm in 1988, the novel object recognition test (NORt), which is a simple and fast procedure to measure recognition memory in rats (*Ennaceur & Delacour, 1988*; *Antunes & Biala, 2012*). Concisely, a rat is allowed to briefly explore a new object introduced in its cage, and, after a temporal interval, the same object is again introduced in the cage, paired with a novel one. Rats tend to ignore the familiar object, and the relative time spent exploring the novel object is taken as a measure of recognition memory. The NOR test is particularly amenable to comparative and development studies, as it is based on the spontaneous tendency towards tactile or visual exploration of novel over familiar stimuli, and the task appears to be less affected by potential confounds due to contingency rules, as well as by potential stress components, due to long training procedures (reviewed in *Blaser & Heyser, 2015*). Researchers have used the NORt paradigm and its variants to study recognition memory in a variety of mammals and birds (*Kornum et al., 2007*; *Ennaceur, 2010*; *Barnes, Burke & Ryan, 2012*; *Soto & Wasserman, 2014*), and more recently they have attempted to use it to study memory in zebrafish using this method (*Gerlai, 2016*). *Lucon-Xiccato & Dadda (2014)* applied a variant of the original NORt to study object recognition in zebrafish. In the familiarization phase, fish were familiarized with one object for 25 min. When subjects were exposed in the test phase to the familiar and a novel object, they spent more time near the novel stimulus. Using similar procedures, other studies found that zebrafish can identify the movement and direction of virtual geometrical shapes (*Braida*

*et al., 2014*) and discriminate objects based on colour and shape (*Oliveira et al., 2015*; *May et al., 2016*) or the previous location of the familiarized object (*Hamilton et al., 2016*). To date, no attempt has been made to study when these important functions emerge and how they develop during ontogeny.

The scope of our study was to determine whether larval zebrafish display some form of recognition memory, when this ability first appears in development and whether there is a reliable procedure to measure it. Most research on zebrafish nervous system, including those on neurodevelopmental and neurodegenerative disorders, is performed in larvae (*Bandmann & Burton, 2010*; *Sager, Bai & Burton, 2010*; *Sakai, Ijaz & Hoffman, 2018*). Larvae can be obtained in large numbers, favouring large-scale screenings of drugs and genotypes (*Norton, 2013*; *Richendrfer et al., 2012*), and their synaptic activation can be monitored in vivo at the whole-brain resolution (*Leung, Wang & Mourrain, 2013*). Among the others, zebrafish models for the re-myelination process in multiple sclerosis and in brain injuries (*Buckley, Goldsmith & Franklin, 2008*), and for alterations in TAU-protein function in pathologies such as Alzheimer's disease (*Paquet et al., 2009*) have been developed in larvae. These and other applications of zebrafish larvae may greatly benefit from efficient methods to measure memory in the early developmental stages.

Zebrafish are generally considered larvae between 3 and 29 dpf. However, before 6 dpf, their visual system does not fully respond to visual stimuli (*Huang & Neuhauss, 2008*). In our experiments, we investigated zebrafish of three ages, which were expected to allow describing developmental trajectories of recognition memory of larvae: 7-, 14- and 21-dpf. Experiments 1 and 2 aimed to assess the ability of larvae to remember familiar objects differing in colour or shape. We adapted the classical NORt paradigm developed for rats (*Ennaceur & Delacour, 1988*). During the familiarization phase, each larva was exposed to two identical objects (experiment 1b: small cubes of the same colour, either red or green; experiment 2: objects of the same colour but with a different shape, cube vs. cone). During the testing phase, the subject was confronted with one familiar object and one of the objects of either a new colour (experiment 1b) or the new shape (experiment 2). We measured the time spent near the familiar or the novel object. Before performing the NOR experiment based on objects with different colour, we assessed the presence of spontaneous colour preferences in larvae. This was done to identify colours that cause innate avoidance or attraction and could impact the results of the NOR experiment. The presence of colour preference is well documented in adult zebrafish (*Spence et al., 2008*; *Avdesh et al., 2012*), but results of studies in larvae are contrasting (*Park et al., 2016*; *Peeters, Moeskops & Veenvliet, 2016*). In experiment 3, we performed a variant of the recognition test using two bi-dimensional printed geometrical figures. Indeed, some studies have adopted bi-dimensional stimuli to study visual recognition in zebrafish and other species (*Braida et al., 2013*; *Braida et al., 2014*) and these stimuli can be easily modified to produce several variants. To better interpret the results of our previous experiments, in experiment 4 we studied ontogenetic changes in the tendency of larvae to approach unfamiliar objects (i.e., neophobia). We positioned subjects in a test apparatus where an unfamiliar object (a black cone) was present and measured the tendency to approach it.
## MATERIAL & METHODS

### Ethical statement

The experiments adhere to the current legislation of our country (Decreto Legislativo 4 Marzo 2014, n. 26) and were approved by the Ethical Committee of University of Padova (OPBA 18/2018, protocol n. 159333 - 30/03/2018).

### Subject

The subjects were wild-type zebrafish larvae of three different ages: 7-, 14- and 21-days post-fertilization (dpf) obtained by spawning from a strain maintained in our laboratory and originally bought in a local pet shop. Larvae were housed in Petri dishes (10 cm Ø, h:1.5 cm) in a solution of Fish Water 1× (0.5 mM NaH2PO4 * H2O, 0.5 mM Na2HPO4 * H2O, 1.5gr Instant Ocean, 1l de-ionized H2O) and Methylene blue (0.0016gr/l) at a density of approximately 30 individuals each. The illumination was set on a 14:10 h light:dark cycle and the temperature was maintained at 28.5 ± 1 °C. Larvae were fed two times a day with dry food (particle size: 0.75 mm) from the age of 6 dpf.

We planned to test 40 zebrafish in experiment 1a for each age (120 larvae in total), 20 in experiment 1b for each age (60 larvae in total), 20 in experiment 2 for each age (60 larvae in total), 20 in experiment 3 for each age (60 larvae in total) and 20 in experiment 4 for each age (60 larvae in total). Forty zebrafish (8 subjects for experiment 1a, 13 subjects for experiment 1b, 6 subjects for experiment 2, 2 subjects for experiment 3 and 11 subjects for experiment 4) were discarded and substituted with new subjects to maintain the predetermined sample size (see details below). The overall study included 360 subjects that completed the four experiments, plus the 40 zebrafish discarded (total: 400).

### Experiment 1a: colour Preference

We used a setup similar to one previously used for studying spontaneous colour preference in adult zebrafish (*Oliveira et al., 2015*). The experimental apparatus (Fig. 1A) consisted of a Petri dish filled with one cm of Fish Water at 28.5 ± 1 °C. To prevent the possible effects of external cues, the apparatus was placed in the centre of an empty, white room with uniform illumination. The water was changed at every trial. The wall and bottom of the petri dish presented LEGO® bricks of four different colours, namely blue (RGB: 0, 61, 165), green (RGB: 0, 173, 69), yellow (RGB: 255, 237, 0) and red (RGB: 227, 0, 11). The LEGO® bricks subdivided the platform into four equivalent sectors. We built the apparatus with LEGO® bricks as the stimuli used for the following four experiments. We used three different colour combinations: (clockwise) red–green–blue–yellow; red–blue–yellow–green; red–yellow–green–blue. The orientation of the colours was also rotated across subjects. At the centre of the platform, a grey plastic square (1 × 1 cm) was used as the starting point during the test. The apparatus was illuminated by a 30-W fluorescent lamp. A Canon LEGRIA HFR38 was positioned at 90 cm above the apparatus for video recording. Each subject was individually inserted in the centre of the apparatus. The behaviour was recorded for 10 min.

### Experiment 1b: novel object recognition test (object's colour)

The experimental apparatus (Fig. 1B) consisted of a Petri dish (10 cm Ø, h:1.5 cm) filled with one cm of Fish Water. The walls of the apparatus were covered with white paper to

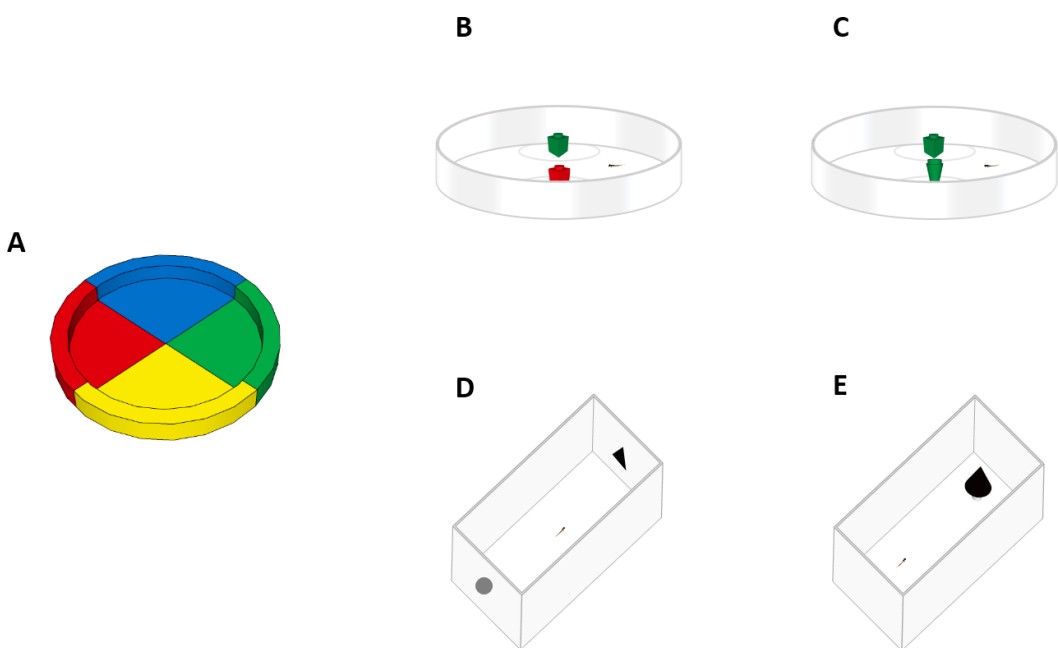

**Figure 1** **Experimental apparatuses.** Top view of the experimental apparatuses used in this study. (A) Experiment 1a (spontaneous colour preference). Larvae were observed in a petri dish subdivided into four equal sectors. (B) Experiment 1b (NOR test). Larvae were familiarized to two objects of the same colour and tested with one familiar object and one of a different colour. (C) Experiment 2 (NOR test). Larvae were familiarized to two objects of the same shape and tested with one familiar object and one of a different shape. (D) Experiment 3 (NOR test). Larvae were collectively familiarized to one printed geometric figure and then individually tested in a rectangular arena with one familiar geometric figure and a novel one. (E) Experiment 4 (neophobic response test). Larvae were placed in a rectangular arena containing an unfamiliar object (a black cone), then we measured the time spent in the vicinity of the object.

prevent the fish from seeing the surrounding room. The apparatus was illuminated by a 30-W fluorescent lamp, and the temperature was maintained at 28.5 ± 1 °C. A Canon LEGRIA HFR38 was positioned at 90 cm for video recording. Stimuli consisted of plastic cubes (1 × 1 cm, LEGO® ID: 3005) of two different colours. Based on the result of experiment 1a, larvae showed a similar preference for green and red sectors built with LEGO® bricks. We chose these two colours for the stimuli. The familiarization phase lasted for 3 days. On the evening of the first day, subjects were individually introduced to the experimental apparatus with the stimuli already present. During the familiarization phase, the two presented objects were identical, either two red or two green LEGO® cubes. The colour of the stimulus was counterbalanced amongst the subjects. Three times a day, with a 4-h interval, both objects were removed for 10 min and then positioned again in the apparatus, in a different position. This changing was aimed to familiarize the subjects to the disappearance and reappearance of the objects. On the morning of the fourth day, subjects were fed 1-h before the test. The test consisted of removing the two identical LEGO® cubes and, after 10 min, replacing them with two LEGO® cubes that differed in colour. We presented one identical LEGO® cube that the subjects were familiar with and a novel one the subjects had not yet experienced. The position of the familiar and novel

stimulus was randomly varied across subjects. The exploratory behaviour was recorded for 8 min.

### Experiment 2: novel object recognition test (object's shape)

Apparatus and procedure were the same for the previous experiment 1b, except for the fact that the stimuli were two LEGO® bricks of the same colour but different shape. They were a green LEGO® cubes (1 × 1 cm; LEGO®, ID: 3005) and a green cone (1 × one cm; LEGO®, ID: 59900; Fig. 1C).

### Experiment 3: recognition of bi-dimensional geometrical figure

We used two different apparatuses for the familiarization phase and the test phase. The apparatus used to habituate the subjects was made of a single Plexiglas tank of 22 × 10 × 12 cm, and it was filled with six cm of Fish Water 1×. Stimuli were black bi-dimensional geometrical figures made with Microsoft PowerPoint and printed on white paper with a laser printer. Stimuli were placed along the outer sides of the short walls of the familiarization tank (Fig. 1D). The long walls of the familiarization tank were covered with white paper to prevent the fish from seeing the room.

The test apparatus (Fig. 1D) consisted of a single Plexiglas tank (8 × 4 × 5 cm) filled with 2.5 cm of Fish Water (80 mL). The temperature was maintained at $28.5 \pm 1$ °C. The bottom and the long sides of the test tank were covered by white paper to prevent external interference. Familiarization and test apparatus were illuminated by a 30-W fluorescent lamp. A Canon LEGRIA HFR38 was positioned at 90 cm above the test apparatus for video recording. The stimuli were presented on the shorter side of the test apparatus. Stimuli consisted of a black circle geometrical figure (0.62 cm $\varnothing$, area: 0.3 cm$^2$) and a black triangle geometrical figure (l: 0.86 cm, h: 0.72 cm, area: 0.3 cm$^2$). The position of the stimuli was counterbalanced between the shorter sides of the test apparatus. The familiarization phase lasted 4 days. In the morning of the first day, subjects were introduced to the familiarization tank where one out of two bi-dimensional geometrical figures were presented (either a circle or a triangle). On the fifth day, each subject was individually inserted in the test apparatus presenting both the familiar geometrical shape and the novel one. The exploratory behaviour was recorded for 12 min.

### Experiment 4: development of Neophobia

The test apparatus (Fig. 1E) consisted of a Plexiglas tank (8 × 4 × 5 cm) filled with 2.5 cm of Fish water 1× (80 mL). The test apparatus was covered with white paper up to four cm. The apparatus was illuminated by two 30-W fluorescent lamps. The stimulus consisted of a black cone (diameter base: 0.7 cm, h: 1 cm) placed over a white pedestal (h: 1.3 cm). The position of the stimulus was counterbalanced across subjects between the shorter sides of the test apparatus. A Canon LEGRIA HFR38 was placed 90 cm above the experimental apparatus for video recording. Subjects were individually inserted at the centre of the unoccupied half of the test apparatus. The exploratory behaviour was recorded for 10 min.

### Analysis of the recordings

We analysed the performance of subjects from the digital recordings played back on a computer screen. The recordings were coded in Advanced Video Codec High Definition

format, at 25 frames per seconds and high resolution (1,920 × 1,080 pixels), allowing precise spatial and temporal resolution of larvae position. To analyse the exploratory behaviour of the stimuli, we virtually considered specific sectors of the experimental apparatus according to each experiment. In experiment 1a, we divided the test apparatus into four equivalent sectors in correspondence with the colour. In experiments 1b and 2, we considered a circular area (Ø: 2.0 cm) around each stimulus. In experiment 3, we divided the apparatus into four equivalent sectors (2 × 4 cm), and we only considered the area close to the stimuli. In experiment 4, we divided the apparatus into two equivalent sectors. We analysed the video recordings using the software BORIS (Behavioral Observation Research Interactive Software; University of Torino, Torino, Italy) by a blind experimenter. The software calculated the time spent in each sector of apparatus. As a measure of preference for the coloured sector (experiment 1a), preference for the novel object (experiments 1b, 2 and 3) or preference for the unfamiliar object (experiment 4), we computed the proportion of time close to the reference stimulus over the total time spent close to both stimuli (experiments 1b, 2 and 3) or sectors (experiments 1a and 4).

## Statistical analysis

We performed the statistical analysis in RStudio version 1.1.383 (RStudio Team (2015). RStudio: integrated Development for R. RStudio, Inc., Boston, MA URL The statistical tests were two-tailed, and the significance threshold was $p = 0.05$. Descriptive statistics are reported as mean ± standard deviation. http://www.rstudio.com/). Data were checked for normality before the analysis using the Kolmogorov–Smirnov test. Because the proportion of time spent in each coloured sectors in experiments 1a did not show a normal distribution, we performed an arcsine square-root transformation. Data presented two outliers (one 21-dpf larvae in experiment 1b, and one 14-dpf larva in experiment 3) consisting of subjects that spent 80% more close to one stimulus due to the less activity compared with the average of the other subjects. Because diagnostic plots showed that these two outliers substantially reduced models' fit, we dropped them from the datasets. We analysed the time spent close to both stimuli and the proportion of time close to the reference stimulus (see details above) using ANOVA to evaluate differences amongst factors. We fit models with the considered independent variables, ages as the fixed factor in all five experiments and the coloured sectors as the fixed factor only in experiment 1a, as well as the stimulus used to the familiarization phase in experiments 1b, 2 and 3. When we found a significant difference between ages, we computed a Tukey post-hoc analysis to evaluate the differences between the levels of this factor. Then, we compared the proportion of time close to the referenced stimulus for each age by performing a one-sample two-tailed $t$-test against the chance level (50%).

In experiments 1a and 4, we analysed the temporal pattern of larvae preferences using a linear mixed-effects model (LMMs, 'lmer' function of the 'lme4' R package) fitted with the minutes as the covariate, age as the fixed factor and subject ID as the random effect to account for repeated measures. For experiment 1a, we also fitted the LMM with the colour of the sectors as the fixed factor. We evaluated the effect of these factors using 'Anova' function of the 'lmerTest' package.
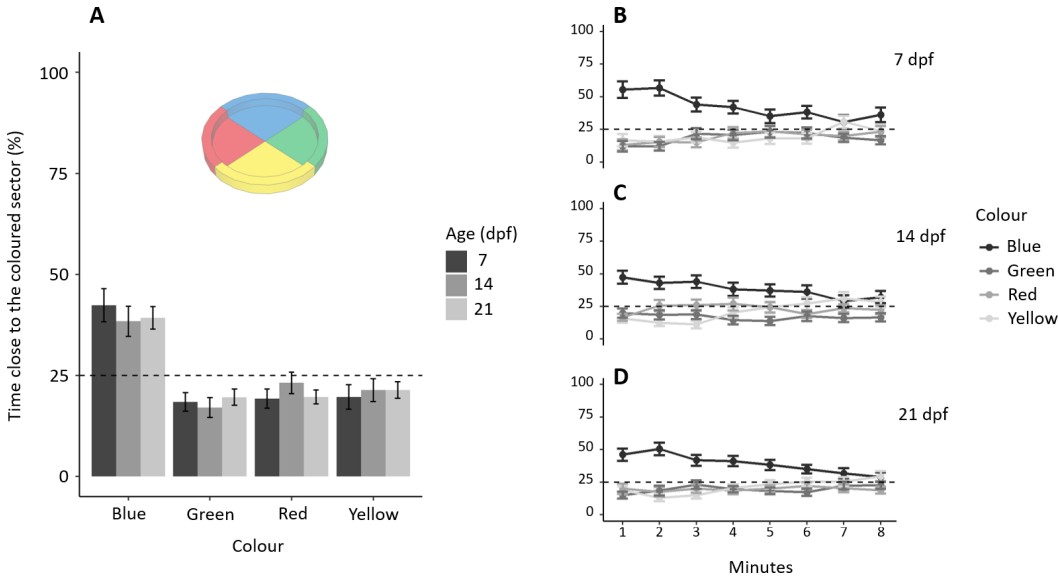

**Figure 2 Colour preference in larvae.** (A) Percentage of time (mean ± standard error) spent in each coloured sector in 7, 14 and 21-dpf larvae. Larvae showed a preference for the blue sectors compared to the other colours (all $P < 0.001$). There was no difference amongst the three ages in the time spent in the four sectors ($P = 0.705$). (B–D) Temporal pattern of time spent in the four sectors for each age. The preference for blue decreased during the trial ($P < 0.001$). Dotted lines represented the expected proportion of time in each sector by chance (25%).

In the meta-analysis of time spent near the stimuli, we initially normalized the data of the three NORt experiments ($Z$-score). The transformed data were then analysed with ANOVA fitted with the age and experiment as factors.

## RESULTS

### Experiment 1a: colour preference

Subjects spent a different amount of time in the four sectors ($F_{3,468} = 35.753$, $p < 0.001$, $\eta p^2 = 0.186$), and there was no difference amongst the three ages ($F_{2,468} = 0.349$, $p = 0.705$, $\eta p^2 = 0.001$), interaction ($F_{6,468} = 0.476$, $p = 0.826$, $\eta p^2 = 0.006$; Fig. 2A). A Tukey post-hoc test indicates that subjects spent significantly more time in the blue sector ($192.01 \pm 107.77$ s, mean ± SD) compared to the green ($88.19 \pm 38.02$ s; $p < 0.001$), red ($99.38 \pm 69.47$ s; $p < 0.001$) and yellow sectors ($99.85 \pm 80.61$ s; $p < 0.001$), but there was no difference amongst the green, red and yellow sectors (all $p > 0.5$). Adding the factor "minutes of the test" to the model reveals that preference for blue significantly decreases with time (Figs. 2B, 2C, 2D): sector ($\chi^2_3 = 418.559$, $p < 0.001$, $\eta p^2 = 0.075$), interaction between "minutes of the test" and sector ($\chi^2_3 = 98.130$, $p < 0.001$, $\eta p^2 = 0.025$), and interaction between age and sector ($\chi^2_6 = 13.312$, $p = 0.038$, $\eta p^2 = 0.003$). No other factors or interactions were significant (all $p > 0.06$).

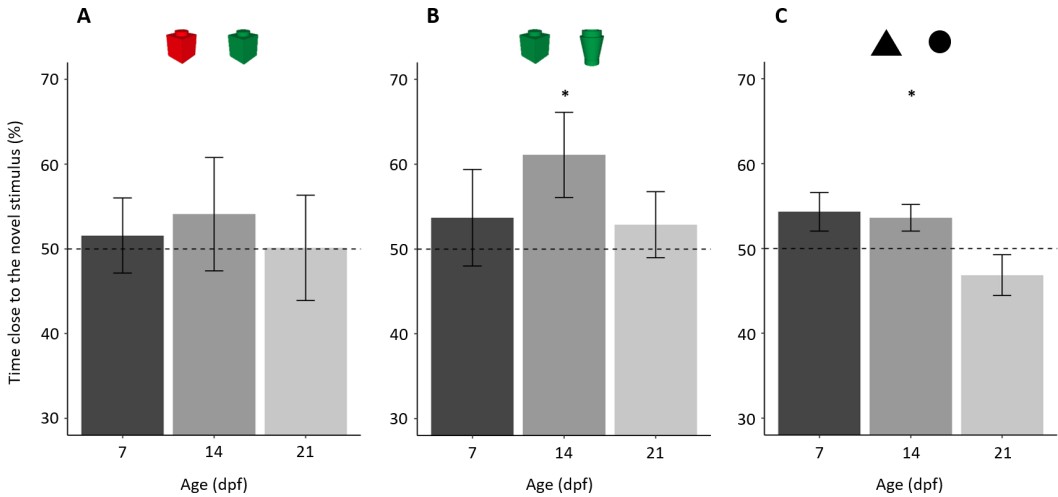

**Figure 3** **Percentage of time (mean ± standard error) close to the novel stimulus in the tree NORt experiments.** (A) Memory for object's colour. Larvae did not show a preference for the familiar or the novel stimulus ($P = 0.559$), and there was no difference amongst the ages ($P = 0.884$). (B) Memory for object's shape. Larvae showed an overall preference for the novel stimulus ($P = 0.043$), and there was no difference amongst the ages ($P = 0.389$). (C) Memory for the shape of a bi-dimensional geometrical figure. Larvae did not show an overall preference for the familiar or the novel stimulus ($P = 0.233$), but the three ages showed a significant difference in preference ($P = 0.032$). Dotted lines represented the expected proportion of time by chance (50%) and asterisks indicated significant differences from chance ($P < 0.05$).

## Experiment 1b: novel object recognition test (object's colour)

In the test phase, subjects, as a whole, spent $139.68 \pm 79.79$ s ($29.10 \pm 16.62\%$) close to one or the other object, and there was no difference amongst the three ages ($F_{2,57} = 3.046$, $p = 0.055$, $\eta p^2 = 0.097$). Larvae did not show a preference for the familiar or the novel stimulus (percentage of time spent close to the novel stimulus $51.96 \pm 25.56\%$; one sample t test: $t_{58} = 0.588$, $p = 0.559$, *Cohen's d* $= 0.077$; Fig. 3A). There was no difference amongst ages ($F_{2,53} = 0.124$, $p = 0.884$, $\eta p^2 = 0.005$) or for the colour of stimulus used during the familiarization phase ($F_{1,53} = 2.511$, $p = 0.119$, $\eta p^2 = 0.045$), interaction ($F_{2,53} = 0.106$, $p = 0.203$, $\eta p^2 = 0.058$).

A separate analysis of the three ages showed that no age group showed a preference for the novel or the familiar stimulus (percentage of time spent close to the novel stimulus in 7-dpf larvae $51.57 \pm 19.86\%$; one sample $t$ test: $t_{19} = 0.353$, $p = 0.728$, *Cohen's d* $= 0.079$; 14-dpf: $54.09 \pm 29.95\%$, $t_{19} = 0.611$, $p = 0.548$, *Cohen's d* $= 0.137$; 21-dpf: $50.12 \pm 27.09\%$, $t_{18} = 0.019$, $p = 0.985$, *Cohen's d* $= 0.004$).

## Experiment 2: novel object recognition test (object's shape)

In the test phase, subjects, as a whole, spent $189.67 \pm 58.75$s ($39.52 \pm 14.41\%$) close to one or the other object, and there was no difference amongst the three ages ($F_{2,57} = 0.767$, $p = 0.469$, $\eta p^2 = 0.026$). Larvae showed a significant preference for the novel stimulus (percentage of time spent close to the novel stimulus: $55.88 \pm 21.98\%$; $t_{59} = 2.073$, $p = 0.043$, *Cohen's d* $= 0.268$, Fig. 3B). There was no difference amongst ages ($F_{2,54} = 0.961$, $p = 0.389$, $\eta p^2 = 0.034$) or in the shape of stimulus used in the familiarization

phase ($F_{1,54}$ = 1.085, $p$ = 0.302, $\eta p^2$ = 0.020); the interaction between these two factors was significant ($F_{2,54}$ = 4.761, $p$ = 0.013, $\eta p^2$ = 0.150). To understand this significant interaction, we performed a split analysis split for age and a split analysis for the stimulus used for familiarization.

The shape of stimulus used for familiarization had a significant effect in 7-dpf larvae ($t_{18}$ = 3.031, $p$ = 0.007, *Cohen's d* = 1.362), but not in 14-dpf subjects ($t_{18}$ = 0.695, $p$ = 0.496, *Cohen's d* = 0.311) or in 21-dpf subjects ($t_{18}$ = 0.635, $p$ = 0.533, *Cohen's d* = 0.286).

A separate analysis of the three ages showed that 14-dpf subjects had a significant preference for the novel stimulus (61.10 ± 22.44%; $t_{19}$ = 2.212, $p$ = 0.039, *Cohen's d* = 0.494), whereas 7-dpf (53.69 ± 25.47%; $t_{19}$ = 0.684, $p$ = 0.525, *Cohen's d* = 0.145) and 21-dpf subjects (53.93 ± 17.23%; $t_{19}$ = 0.733, $p$ = 0.472, *Cohen's d* = 0.164) did not.

## Experiment 3: recognition of bi-dimensional geometrical figures

In the test phase, subjects, as a whole, spent 303.04 ± 110.65 s (42.09 ± 15.37%) close to one or the other stimulus with a significant difference amongst ages ($F_{2,56}$ = 14.482, $p$ < 0.001, $\eta p^2$ = 0.341). A linear trend analysis showed that time spent close to the stimuli significantly decreased with age ($p$ < 0.001). Larvae did not show a preference for the familiar or the novel stimulus (percentage time spent close the novel stimulus: 51.56 ± 9.94%; $t_{58}$ = 1.205, $p$ = 0.233, *Cohen's d* = 0.157; Fig. 3C). The ANOVA on percentage time spent close to the familiar stimulus found a significant effect of age ($F_{2,53}$ = 3.691, $p$ = 0.032, $\eta p^2$ = 0.122). There was no significant effect of stimulus used for familiarization ($F_{1,53}$ = 1.940, $p$ = 0.169, $\eta p^2$ = 0.035) nor significant interaction ($F_{2,54}$ = 0.479, $p$ = 0.622, $\eta p^2$ = 0.017).

A separate analysis of the three ages showed that 14-dpf subjects had a significant preference for the novel stimulus (53.61 ± 7.08%, $t_{18}$ = 2.224, $p$ = 0.039, *Cohen's d* = 0.510). Seven-dpf and 21-dpf subjects did not show significant preference (7-dpf: 54.32 ± 10.16%, $t_{19}$ = 1.902, $p$ = 0.073, *Cohen's d* = 0.425; 21-dpf: 46.85 ± 10.74%, $t_{19}$ = 1.313, $p$ = 0.205, *Cohen's d* = 0.294). However, the effect size and the $p$ value close to the threshold for statistical significance suggest that the 7-dpf subjects might have showed a trend of preference similar to that of 14-dpf subjects.

## Meta-analysis of the three NOR experiments
### Overall analysis of the three ages
Despite the presence of minor differences, the three experiments essentially measured the same parameter (i.e., the tendency to share time between a familiar object and a novel one). We performed a global analysis of the three NORt experiments at the three ages.

*Time spent near the stimuli.* Larvae showed a general tendency to decrease the time close to the stimuli with age ($F_{2,169}$ = 13.970, $p$ < 0.001, $\eta p^2$ = 0.142; linear trend: $p$ = 0.036; Fig. 4). There was a significant difference amongst the three experiments ($F_{2,169}$ = 64.078, $p$ < 0.001, $\eta p^2$ = 0.431) and interaction ($F_{4,169}$ = 3.921, $p$ = 0.005, $\eta p^2$ = 0.085).

*Preference for the novel stimulus.* Overall, larvae showed a preference for novel stimulus (53.15 ± 20.28%; $t_{177}$ = 2.071, $p$ = 0.040, *Cohen's d* = 0.155). There was no effect of age

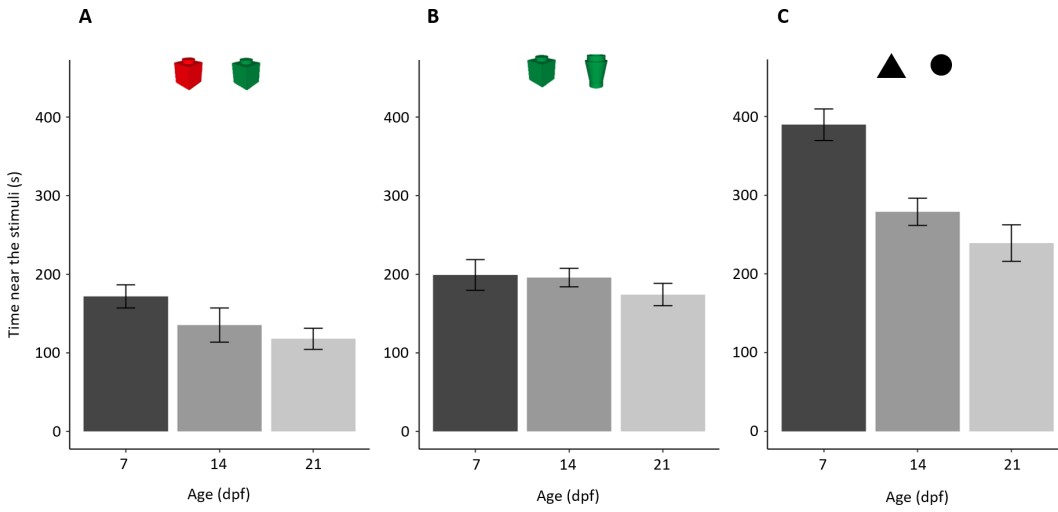

**Figure 4  Time (mean ± standard error) spent close to both stimuli in the three NORt experiments.**
Overall, time close to the stimuli decreased with age ($P < 0.001$; linear trend: $P = 0.036$) with a significant difference amongst experiments ($P < 0.001$) and interaction.

($F_{2,69} = 1.438$, $p = 0.240$, $\eta p^2 = 0.017$), experiment ($F_{2,169} = 0.814$, $p = 0.445$, $\eta p^2 = 0.010$) or significant interaction ($F_{4,169} = 0.277$, $p = 0.893$, $\eta p^2 = 0.007$).

### Analysis of 14-dpf larvae

Various lines of evidence point to the possibility that only 14-dpf larvae fully responded to the NORt paradigm (see discussion). We performed a joint analysis of the three NORt experiments restricted to larvae of this age.

*Preference for the novel stimulus.*  We found a significant preference for the novel stimulus ($56.31 \pm 22.05\%$; $t_{58} = 4.635$, $p < 0.001$, $\eta p^2 = 0.603$). There was no significant difference in the preference for the novel stimulus amongst the three experiments ($F_{2,56} = 0.707$, $p = 0.497$, $\eta p^2 = 0.025$).

*Effect of stimulus used in the familiarization phase.*  There was a significant effect of this factor in experiment 1b ($t_{18} = 2.194$, $p = 0.042$, *Cohen's d* $= 0.986$), indicating a spontaneous preference of 14-dpf larvae for the red over the green colour (Fig. 5A). No difference between stimuli was found in the other two experiments (experiment 2: $t_{18} = 0.695$, $p = 0.496$, *Cohen's d* $= 0.311$, Fig. 5B; experiment 3: $t_{17} = 1.532$, $p = 0.144$, *Cohen's d* $= 0.704$, Fig. 5C).

## Experiment 4: development of Neophobia

In the test phase, subjects, as a whole, spent $208.16 \pm 133.31$ s ($34.69 \pm 22.22\%$) close to the novel stimulus with a significant difference amongst ages ($F_{2,57} = 3.198$, $p = 0.048$, $\eta p^2 = 0.101$; Fig. 6A). A significant linear trend ($p = 0.015$) indicates that the proportion of time close to the stimulus decreased with increasing age. The proportion of time close to the stimulus increased throughout the test ("minutes of the test": $\chi^2_1 = 34.973$, $p < 0.001$,

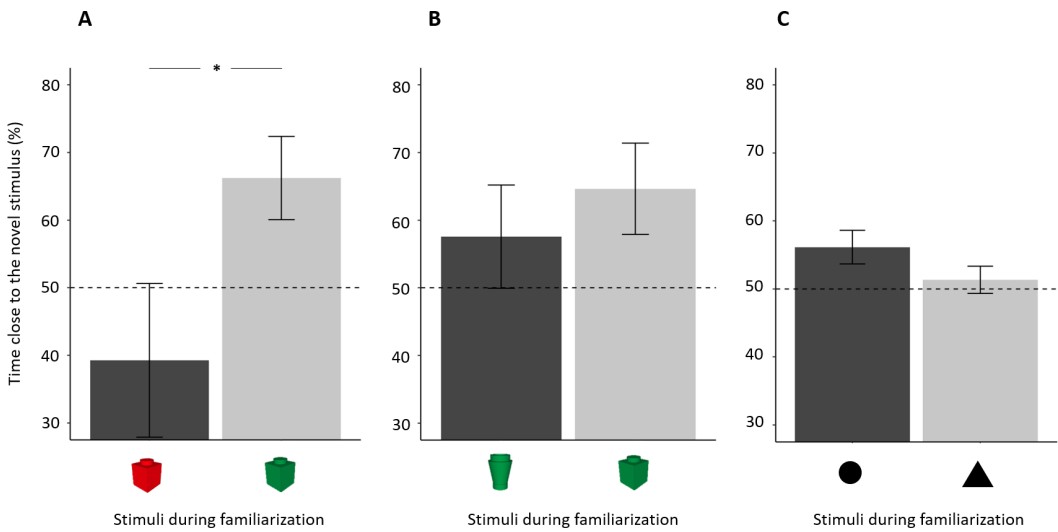

**Figure 5** Percentage of time (mean ± standard error) close to the novel stimulus in 14-dpf larvae in relation to the stimulus used in the familiarization phase. (A) Subjects showed a spontaneous preference for the red colour, and they tended to prefer the red colour for both familiar and novel objects. (B), (C) No difference was found in the two other NORt experiments. Dotted lines represented the expected proportion of time by chance (50%) and asterisks indicated significant differences between the two conditions ($P < 0.05$).

$\eta p^2 = 0.061$; linear trend: $p < 0.001$) with a significant difference amongst ages ($\chi^2_2 = 6.341$, $p = 0.042$, $\eta p^2 = 0.014$), but not the interaction ($\chi^2_2 = 4.814$, $p = 0.090$, $\eta p^2 = 0.009$). A separate linear trend analysis for the three age groups shows that larvae of all the three ages significantly decreased their neophobia during the test (7 dpf: $p = 0.004$; 14 dpf: $p = 0.049$; 21 dpf: $p < 0.001$; Figs. 6B–6D).

## DISCUSSION

To assess the presence of recognition memory in zebrafish larvae, we adapted the most used paradigm in this field, the Novel Object Recognition tests (NORt), a procedure that exploits the tendency of most vertebrates to explore objects they have never seen before (*Ennaceur, 2010*). We observed an overall tendency of larvae to spend more time in the vicinity of the novel compared to the familiar stimulus, indicating that recognition memory likely emerges in zebrafish from the first weeks of life. However, the preference appears to be fully significant only in the 14-dpf larvae and in two out of three recognition experiments, suggesting that subjects' age and type of stimuli may affect memory assessment.

The non-linear effect of age in the recognition memory experiments (i.e., 14-dpf larvae >7-dpf larvae = 21-dpf larvae) was likely due to two concomitant causes. The first is an ontogenetic change in the propensity to approach a novel object, which is commonly observed in many species because of experience, maturation or age-specific variation in ecology (*Kendal, Coe & Laland, 2005*; *Miller et al., 2015*). Often, young individuals tend to explore all new objects, and neophobic response increases as they grow older (*Menzel, 1966*; *Biondi, Bó & Vassallo, 2010*). Researchers have suggested that animals begin their

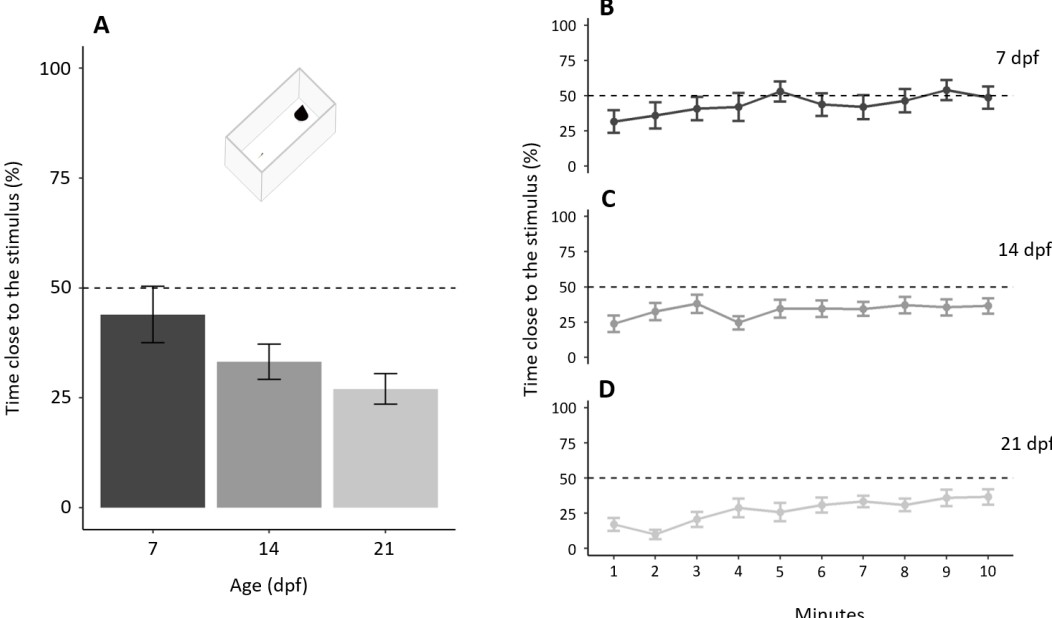

**Figure 6  Development of neophobia in larvae.** (A) Time near a new stimulus decreases with increasing age ($P = 0.048$; linear trend: $P < 0.001$). (B–D) Tendency to approach the stimulus for each age increased throughout the test ($P < 0.001$). Dotted lines represented the expected proportion of time by chance (25%).

life with almost no information about their environment and that the potential benefits of exploring new objects are high (*Shettleworth, 2010*); such benefits decrease, as the experience accumulated and the costs associated with approaching unfamiliar objects may prevail in older individuals (*Greenberg & Mettke-Hofmann, 2001*). As shown by experiment 4, this latter pattern seems to characterize zebrafish as well. When exposed to an unfamiliar object in an arena, larvae increased their neophobia with age and time spent in close vicinity of the unfamiliar object by 21-dpf larvae as almost half that spent by 7-dpf larvae. Therefore, it is possible that, in our experiments, 21-dpf larvae do discriminate the novel from the familiar stimulus, but their exploration tendency was counterbalanced by an increasing neophobia that hindered the detection of recognition memory. In other species, the neophobic reaction may prevail. Miletto Petrazzini and colleagues (*2012*), testing 5-day old guppies, observed an initial neophobic response to a novel object introduced in their home tank. Tested with a NOR procedure, young guppies spent significantly more time near a familiar object than a novel one. The finding that 21-dpf larvae, as well as larvae from the other age groups, showed a neophobic response toward an unfamiliar object in experiment 4 may reinforce the idea that they possess some form of recognition memory. Various authors have observed that to recognize that an object is new, an animal needs to recall features of previously encountered objects (*Hughes, 2007*; *Greggor, Thornton & Clayton, 2015*). However, it is difficult to demonstrate this involvement of recognition memory in experiment 4. Subjects did not experience objects before the test and, without

the control stimulus, it cannot be excluded that they showed a generalized neophobia against any stimulus.

The second possible cause of the age effect is the development of neural circuits that support recognition memory or visual discrimination in general. Differing from many other species of fish, the zebrafish shows a very rapid embryonic development with only three days occurring from fertilization to hatching. Therefore, at birth, the brain of zebrafish larvae is in a very immature stage of development (*Nusslein-Volhard & Dahm, 2002*). Larvae start to feed autonomously only at 6 dpf, whereas more complex functions such as sociality appear much later in development (*Roberts, Bill & Glanzman, 2013*; *Dreosti et al., 2015*). The poor response of 7-dpf larvae to the recognition memory tests may derive from the fact that neural structures crucial to visual discrimination and recognition memory are relatively undeveloped or developed in some individuals but not in others. This cognitive effect remains to be addressed because to date there are no other tests available to measure recognition memory in zebrafish larvae.

The difference between recognition experiments was likely due to the presence of innate preference/avoidance towards some of the object's features. Biases in novelty responses have been documented in a variety of organisms (*Fantz, 1957*; *Dorosheva, Yakovlev & Reznikova, 2011*) and could intuitively affect a measure of recognition memory based on the relative preference in approaching two objects. This factor could be particularly important in our study given the early age of the subjects and their poor perceptual experience, due to the maintenance in a bare petri dish as required by the standard laboratory housing conditions for zebrafish larvae. The influence of such a factor is evident in experiment 1b. Larvae familiarized to green cubes tend to prefer the novel colour (red cube), whereas larvae familiarized to red cubes show a preference for the familiar colour (red cube). Since colour preferences were previously observed in both adults and larval zebrafish (*Oliveira et al., 2015*; *Peeters, Moeskops & Veenvliet, 2016*), before the experiment, we had assessed the colour preference in larvae. Fish of all three ages showed a consistent attraction to the blue colour, whereas the other three colours (green, red and yellow) seemed to be similarly preferred. In light of the results of experiment 1b, it is likely that larvae have an innate preference of red over green and that this preference was masked in experiment 1a by the strong attraction to blue. Other studies have suggested another methodological factor that is critical for recognition memory and could explain the difference between experiments in our study, that is the similarity between stimuli (e.g., *Bettis & Jacobs, 2012*). Yet, it seems difficult to attribute our results to this factor. Indeed, we found no evidence of recognition in the experiment with colour stimuli, but in the preference test of experiment 1a, zebrafish proved able to distinguish between colours. It remains to be addressed whether the use of objects and images that differ more would improve the recognition performance of the younger group of zebrafish (7 dpf).

Overall, our results confirm that recognition memory can be assessed in zebrafish larvae, provided that subjects' age and type of stimuli are carefully evaluated. This conclusion is consistent with research on other species. The NORt has generally been considered a robust test to measure recognition memory in rodent species (*Antunes & Biala, 2012*). However, researchers also reported various limitations, mainly due to the influence of

non-cognitive factors (*Ennaceur, 2010*). For example, besides memory, the NORt is likely affected by the individual propensity to approach a novel object, which affects the amount of information about the objects acquired during familiarization, as well as the measure of preference in the test phase (*Akkerman et al., 2012*). Several methodological details, such as trial duration and previous experience of the subjects, might also affect the NORt's results (*Dere, Huston & Silva, 2007*). The results of fish experiments have revealed similar contrasting effects. For example, a recent study on sex differences in guppies found that males explored the novel object at the beginning of the experiment, whereas females did so at the end (*Lucon-Xiccato & Dadda, 2016*). According to the temporal windows considered, one sex or the other would appear to perform more, but an overall analysis revealed no sex difference in recognition memory. Contrary to other studies (*Braida et al., 2014*; *Lucon-Xiccato & Dadda, 2014*; *Oliveira et al., 2015*), May and colleagues (*2016*) found that zebrafish preferentially approached familiar over novel objects and that this response was further modulated by the size of the objects.

The zebrafish is rapidly gaining ground as a model for brain diseases due to great ease in dissecting the genetic and physiological basis of these pathologies very early, in some cases even during embryonic development or in the first days of life (*Buckley, Goldsmith & Franklin, 2008*; *Leung, Wang & Mourrain, 2013*; *Paquet et al., 2009*; *Spence et al., 2008*). Although improvements are still needed, the recognition memory task that we developed may provide an important tool to assess early cognitive functioning zebrafish. Researchers on neuropathologies could enjoy the advantage of detecting memory dysfunctions in their subjects at the age of 14 dpf without waiting to test adult fish (*Lucon-Xiccato & Dadda, 2014*; *May et al., 2016*). In addition, many mutagenic lines show high mortality, reducing the number of available adult subjects. The simplicity and low costs of the NORt for larvae may allow using this method as an initial screening of large populations. An additional point of interest regards animal welfare because the NORt is based on spontaneous behaviour and does not require harmful manipulations.

## CONCLUSIONS

We investigated the presence of recognition memory, the developmental evolution of this ability, and the use of a novel object recognition procedure to measure it in zebrafish larvae. Although our experiments generally suggest that zebrafish larvae already possess some form of recognition memory and that this can be measured at 14 dpf, we demonstrated that NOR tests have some limitations in assessing it. In fact, at least in the version developed for rats, this test seems influenced by non-cognitive factors, such as neophobia, previous experience and spontaneous preferences. Therefore, we need more studies pursuing the objective of devising simple procedures to measure recognition memory in zebrafish larvae.

## ACKNOWLEDGEMENTS

The authors would like to thank Carola Zoboli, Lorenzo Esposito, Jessica Gatto, and Margot Carli for their help in testing the animals, and Stefano Massaccesi from Department of General Psychology for his help in building the experimental apparatus. This work was

carried out within the scope of the project "use-inspired basic research", for which the Department of General Psychology of the University of Padova has been recognized as "Dipartimento di eccellenza" by the Ministry of University and Research (MIUR).

### Funding

This research was supported by DOR Grant to Angelo Bisazza from the University of Padova. The funders had no role in study design, data collection and analysis, decision to publish, or preparation of the manuscript.

### Grant Disclosures

The following grant information was disclosed by the authors:
University of Padova.

### Competing Interests

The authors declare there are no competing interests.

### Author Contributions

- Matteo Bruzzone conceived and designed the experiments, performed the experiments, authored or reviewed drafts of the paper, and approved the final draft.
- Elia Gatto analyzed the data, prepared figures and/or tables, authored or reviewed drafts of the paper, and approved the final draft.
- Tyrone Lucon Xiccato conceived and designed the experiments, analyzed the data, authored or reviewed drafts of the paper, and approved the final draft.
- Luisa Dalla Valle and Angelo Bisazza conceived and designed the experiments, authored or reviewed drafts of the paper, and approved the final draft.
- Camilla Maria Fontana and Giacomo Meneghetti performed the experiments, authored or reviewed drafts of the paper, and approved the final draft.

### Animal Ethics

The following information was supplied relating to ethical approvals (i.e., approving body and any reference numbers):

The experiments adhere to the current legislation of our country (Decreto Legislativo 4 Marzo 2014, n. 26) and were approved by the Ethical Committee of University of Padova (OPBA 18/2018, protocol n. 159333 - 30/03/2018).

### Data Availability

The raw measurements are available in the Supplemental Files.

### Supplemental Information

Supplemental information for this article can be found online at http://dx.doi.org/10.7717/peerj.8890#supplemental-information.

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
