# Peer review of "Measuring recognition memory in zebrafish larvae: issues and limitations"

_PeerJ, doi:10.7717/peerj.8890_

## Round 0.1 · original submission · Minor Revisions

Dear Sirs:
Please review carefully and answer the comments of both reviewers, but especially those of reviewer #1, in a revised version of your manuscript.

Reviewer 1 ·

Basic reporting

The article provides sufficient introduction and literature references, however, I suggest some changes listed in the general comments below.

Experimental design

The experiments are well performed, however I have some concerns regarding the relevance of the study.

Validity of the findings

I suggest some changes in the Discussion and Conclusions sections. Please read the general comments for the author.

Additional comments

In this paper, Bruzzone et al analyze recognition memory in zebrafish larvae. The main conclusion of the study is that zebrafish larvae presents recognition memory, but some “non-cognitive” factors may complicate the assessment or interpretations of the tests. I found some difficulties understanding the relevance of the study and I have some concerns about the experimental design and the interpretation of the data.

While reading the abstract and introduction, I could not find a clear statement justifying the study. Authors list various papers from previous literature where recognition memory (NOR) has been found in zebrafish, but a clear explanation on the importance of assessing this memory in early stages of development must be provided.

In the introduction, I found the nomenclature regarding the developmental stages confusing. For the reader to have a quick and efficient comparison with previous reports, expressions as “third week of life” must be substituted by clear periods or days. To give consistency with the results section I would recommend changing everything to days post fertilization.

From line 53 to 61 in the introduction, authors listed a couple of papers where zebrafish are place-conditioned with an electroshock or pairing tail contractions with spotlights. The authors draw a justification for the present study by questioning the possibility that larvae can learn a “more advance” form of learning and memory. The claim that recognition memory could be considered a more advance form of learning must be logically justified and references for this must be provided. I suggest avoiding unclear concepts, especially on the justification of the study.

Please provide references for the statements on lines 69 to 72, on the advantages of using object recognition memory.

Experimental design and methods:
Please provide a clear justification for the selected post-fertilization days.

Please provide image resolution and sampling rates used in video recordings and provide a clear explanation on how these parameters are appropriate to track, with sufficient spatial and temporal resolution, the movement of the larvae.

Please provide a rational explanation to use two different novel object recognition tests (3D, Fig. 1A-B vs 2D Fig. 1D). In fact, I would be tempted not to consider the “Bi-dimensional” version as an object recognition test. This would be more similar to a place preference test, where different cues in the environment bias the preference of the subjects for one or other places. In this experimental design, subjects don’t have access to directly interact with the object. This conceptual clarification is necessary to avoid confusion in the interpretation of the data.

Some clarifications on the design of the experiment are needed. First, the dimensions of the box are different than the other experiments. This one is a smaller box; the reason of this change must be justified. Is it possible that as animals grow, they would feel less comfortable in a smaller environment?

Results

For all figures, asterisks indicating significant differences should be included.
In Experiment 1b, the preference index reported 55.88% + 21.98%, this is a very low index of preference, does that mean that not all the animals presented preference over the novel object? The use of parametric statistics and representations makes it even more difficult to understand. Is the data normally distributed? Please clarify and justify the use of parametric statics and representations for this and the rest of the data.
In line 293, the phrasing “marginally significant tendency” is not clear, in parenthesis the p = 0.073 indicates not significance.
In the meta-analysis (lines 297 to 309), please refer every subpanel of figure 5 in the text.

Discussion

I suggest addressing the possibility that the objects used in the study are not sufficiently different from each other.
Discussion must include a section illustrating the relevance of the main finding, that is, zebrafish larvae display higher levels of object recognition in the 14-dpf. How this data advances our knowledge regarding learning and memory? Please provide a clear statement on the pertinence of keep using this model in these developmental stages.
The findings of neophobia are in my opinion very interesting. Neophobia may represent a “complex cognitive” feature that may require, among other things, memory. For example, wouldn’t neophobia be a sort of object recognition test? How come an animal can avoid something new if it does not classify it as something new? I would suggest a few lines on the significance of this results and not only using it as a justification for the lack of object recognition after 21-dpf.
Conclusion must address the three objectives of the study (lines 85 to 87), on whether zebrafish larvae display recognition memory, when this ability first appear in development and whether there’s a reliable method to asses it. Speculations on neurodegenerative disorders should be part of the discussion.

Reviewer 2 ·

Basic reporting

See General Comments for the Authors.

Experimental design

See General Comments for the Authors.

Validity of the findings

See General Comments for the Authors.

Additional comments

The authors performed behavioral tests on zebrafish larvae to test if, and at which age, the larvae develop object recognition capabilities. Larvae of 7, 14 and 21 days post-fertilization were tested on three variants of the Novel Object Recognition test (NORt). The results show that, overall larvae explored the novel object 53% percent of the time, a small but statistically significan effect. The results also the novel object recognition test peaks at age 14 dpf and decreases by 21 dpf, an effect that might be due to the behavior of neophobia developing at later stages.

Overall, the experiments and behavioral measurements seem well performed and the manuscript is adequately structured and written.

I have only one major question. Did authors rotate the location of the colors (and forms) so that, for example, blue could to the north but also south and west and east from the center of the petty dish/tank? This is important since preference for color of form might be confused by other environmental factors that made the larvae swim to a particular location of the petri dish or tank. Thus, is it important that the authors randomly change (or counterbalance) the spatial orientation of the behavioral apparatuses.

Minor comments:

-Line 26. Remove “that”.

-Line 96. The phrase needs rewording.

-Line 229. Define “IS”.

-In figure 3, please use lines and asterisks to illustrate the significant differences.

-Figure 5 show a preference for the green color, as opposed to the red preference mentioned in the text. Are labels wrong on the figure?

---

## Round 0.2 · accepted · Accept

Your article has been approved for publication.

Reviewer 1 ·

Basic reporting

No comment

Experimental design

No comment

Validity of the findings

No comment

Additional comments

The authors addressed all my previous concerns.

Reviewer 2 ·

Basic reporting

No comment.

Experimental design

No comment.

Validity of the findings

No comment.

Additional comments

The authors addressed my comments and queries satisfactorily. Thank you.